# Performance Level and Strike Type during Ground and Pound Determine Impact Characteristics and Net Force Variability

**DOI:** 10.3390/sports10120205

**Published:** 2022-12-13

**Authors:** Vaclav Beranek, Petr Stastny, Vit Novacek, Kajetan J. Słomka, Dan Cleather

**Affiliations:** 1Department of Rehabilitation Fields, Faculty of Health Care Studies, University of West Bohemia, 30100 Pilsen, Czech Republic; 2Department of Sport Games, Faculty of Physical Education and Sport, Charles University, 16252 Prague, Czech Republic; 3Biomechanical Human Body Models, New Technologies—Research Centre, University of West Bohemia, 30100 Pilsen, Czech Republic; 4Department of Motor Behavior, Institute of Sport Sciences, The Jerzy Kukuczka Academy of Physical Education in Katowice, 40-002 Katowice, Poland; 5Faculty of Sport, Applied Health and Performance Sciences, St. Mary’s University, Waldegrave Road, London TW1 4SX, UK

**Keywords:** direct punch, elbow strike, palm strike, self-defence, mix martial art

## Abstract

The evaluation of strike impact is important for optimal training, conditioning and tactical use. Therefore, the aim of this study was to evaluate ground and pound strikes, in terms of net force variability, across genders and performance levels. Eighty-one participants, professional men (*n* = 8, 37 ± 6 years, 195 ± 7 cm, 113 ± 27 kg), advanced men (*n* = 47, 26 ± 8 years, 180 ± 7 cm, 76 ± 11 kg), and advanced women (*n* = 26, 21 ± 1 years, 167 ± 6 cm, 61 ± 7 kg) performed three strikes from a kneeling position into a force plate on the ground. The elbow strike resulted in the highest impulse and the palm strike in the highest peak force for all three categories. These results support the recommendation that has previously been made to teach the palm strike to beginners and advanced tactical and combat athletes. The direct punch and elbow strike net force were characterized by a double peak curve, where the first peak variability explained 70.2–84% of the net force. The second peak was pronounced in professional men during elbow strikes, which explained 16% of net force variability. The strike type determines the impact net force and its characteristics, where palm strike is typical by highest peak impact tolerance and elbow strike by double force peak with high net force impulse.

## 1. Introduction

The evaluation of strike impact is essential for understanding the training and conditioning demands of tactical actions in sports, self-defence, and combat [1,2,3]. Although there are several characteristics of the strike that can be considered, such as the kinematics, accuracy, speed of the strike [4,5,6], strike timing, or stiffening of the striking body part [7], the dynamics of the strike’s impact are output parameters that are important in determining the effectiveness of the strike [8,9]. One way of estimating strike output is the strike net force production magnitude; however, there is always a question of what the repeatability of such a force interaction for selected strikes is. 

Strike selection depends on tactical and individual needs, which are in turn influenced by the striker’s anthropometry, strength, mobility, and performance level [10,11]. For instance, the striker’s impact increases with bodyweight, a technique [7,11,12], and years of striking experience [13]. Strikes can differ in terms of their velocity [8,14], the range over which they can be delivered [10] and their impact [4,5]. The choice of strike is dependent on the tactical situation, where the direct punch might be chosen as the velocity of delivery is faster than the opponent’s reaction [15], or alternatively to achieve high peak net force (F_peak_) [16]. For instance, F_peak_ has been shown to be higher in the palm strike than in the straight punch or elbow strike, and to be much lower for women than men [5]. This has been found in the case of the mount position (kneeling position during groundwork) [17], where the striking athlete in mixed martial arts (MMA) sits on the opponent employing any variation of upper limb strikes targeting most often the opponent’s head because hitting the head area is a determining factor for success in MMA [18,19,20]. Despite this, it remains unclear which strikes are more or less appropriate for strikers of different performance or experience levels, and which strikes can be performed with high or low variability.

Although previous research has found there to be a practical meaning to discrete measures of a strike’s impact qualities [8,21], such as in the association between F_peak_ and F_mean_ with victory in a boxing match [16], simplifying the net force expression to just one numerical value means neglecting a large part of the net force production dataset. During impact, the highest F_peak_ does not necessarily equate to the highest energy transfer or impulse expression. However, some level of F_peak_ is required to cause damage to the target, while some strikes might have more force peaks during the contact phase. The contact area and the precision of the striking technique might influence this. Therefore, it is advantageous to describe each strike in terms of the full time-series of the impact forces, rather than just isolated quantities such as F_peak_ or impulse. Moreover, the strike output variability is important as a description of its repeatability and possibly skill level [22,23,24]. In the variability evaluation, the less-skilled individuals exhibit greater variation in performance [25], and it is expected that more difficult movement patterns would result in higher variability, which can be described by principal component analyses (PCA) [26,27]. 

Previous research has typically evaluated the kinematics [28], ground reaction force [29,30,31], net impact force [4,32], mixed designs [33,34,35], or the evaluation of selected properties [14,36] of strikes by comparing discrete values such as peaks or means rather than studying the full time-series and its variability. Therefore, this study aimed to evaluate the time series of net force of three strikes used in ‘ground and pound’ position in the mixed martial arts system (the palm strike, direct punch and elbow strike in a dominant position) across gender and performance levels. We hypothesised that there would be less variability in the force application during the palm strike, and this would not differ between genders or performance levels. 

## 2. Materials and Methods

This experiment used a randomized cross-sectional design, where each participant was familiarized with the testing procedure and the striking actions. The familiarization session was performed 48 h prior to the main testing day. The procedure consisted of a general warm-up of 10 min jogging and stretching with supervised bodyweight exercises followed by a specific warm-up. The specific warm up included a minimum of 15 strikes of varying intensity into the force plate. Each participant consecutively performed 5 straight punches with a clenched fist, 5 palm strikes (straight strikes with an open palm) and 5 elbow strikes (straight strikes using the olecranon of the elbow for short distance) in a randomized order of strike type. Data collection was performed in a biomechanical laboratory by the same investigator and at the same time of day as for the familiarization. 

### 2.1. Participants

The study sample consisted of 81 participants (men and women) in two performance categories, that is, professional men (*n* = 8, 37 ± 6 years, 195 ± 7 cm, 113 ± 27 kg), advanced men (*n* = 47, 26 ± 8 years, 180 ± 7 cm, 76 ± 11 kg), and advanced women (*n* = 26, 21 ± 1 years, 167 ± 6 cm, 61 ± 7 kg). Participants had been practicing self-defence at an advanced (<5 years’ experience) or professional (>5 years’ experience including experiences as instructors) level, were aged 18 years or older, had no injuries or other medical restrictions, and gave signed informed consent. The study protocol was approved by the local ethical committee (No. 267/2019) and was conducted in accordance with the Declaration of Helsinki (2013).

### 2.2. Striking Procedures

Strikes were performed without any protective equipment on the participants’ hands using the dominant hand (there was protection on the testing device). The position of the force plate was adjusted to allow strikes to be executed in an approximately perpendicular direction relative to the force plate which was oriented horizontally in front of the athlete (Figure 1). The strikes were performed from a kneeling position with a 15-s rest interval between strikes of the same type and a 5-min rest between each different type of strike. The athletes executed 5 straight punches with the phalanges of the clenched fist, 5 palm strikes with the metacarpal area of an open palm, and 5 straight elbow strikes using the olecranon in a randomized order of strike type. Participants were instructed to perform all strikes with the maximal effort and impact that they felt comfortable with. 

The initial position for each strike was standardised—kneeling with the participant’s striking hand in contact with their lower jaw (standard defensive “shield” position). The participant then extended the upper limb keeping their body above the plate before the strike and avoiding excessive hip flexion (above 40°), then performed the strike using the selected technique. The starting distance from the force plate was the length of the stretched upper limb in each attempt with the knees 10 cm from the edge of the plate. The athletes were instructed not to touch the measuring device with any part of the body apart from the striking hand during the experiment. If a participant did not comply with the specified measurement protocol, the trial was repeated.

### 2.3. Instrumentation and Data Acquisition

A three-axis force portable plate (Kistler 9286B, Kistler Inc instrumente, GmbH, Winterthur, Switzerland, 600 × 400 × 35 mm) was placed on the floor and the total height of the contact surface was 53 mm. Data was acquired at a sampling frequency of either 1000 Hz or 10,000 Hz (in the latter case, the data was down sampled to 1000 Hz for analysis). The force plate surface was covered with a dense polyethylene of 1.8 cm thickness (Tatami Trocellen). The plate hardness was determined using a durometer (type A, DIN 53505; ASTM D 2240; ISO 7619) where the relative dynamic attenuation was 28–42% in peak force.

### 2.4. Data Curation and Statistical Approach

The Kruskal–Wallis ANOVA was used to compare peak force differences between groups separately in each strike, and the Friedman ANOVA was used to compare peak forces between strikes separately in study groups due to the non-parametric data and different group sample sizes. Effect size was estimated for the Kruskal–Wallis test by eta square (η^2^) calculated from test H value, and for the Friedman ANOVA by e Kendall’s W. All tests were run with α = 0.05 and statistical significance was set up at *p* < 0.05.

The beginning of a strike was identified by finding the point at which the force increased to 300 N above the baseline force. A period of 0.05 s from the beginning of the strike was chosen for analysis (although in the results here only the first 0.02 s is shown). Principal component analysis (PCA) was performed separately on the force-time curves of each group for each strike—that is, 9 separate PCAs were performed. The first 3 principal components (PCs) were retained for further analysis as this was the minimum number of PCs that explained at least 90% of the variance in the dataset for all groups and strikes.

In this study, two of the outputs from the PCA were of most interest: (1) the PC scores which are the time-series of the value of each PC; and (2) the loading coefficient matrix which is the rotation matrix which describes the transformation of the raw data into the coordinate system defined by the PCs. In interpreting the results, we rely upon the fact that any of the raw data trials can be reconstructed from a linear combination of the PC scores multiplied by the relevant loading coefficient. In particular, we reconstructed a representative force-time curve for each group and strike that was based on just the first 3 PCs and the mean values of the loading coefficients. Finally, the impulse for each strike was calculated by taking the area underneath the reconstructed force-time curve. All analysis was performed using MATLAB^®^ (R2020a; The Mathworks Inc., 1 Apple Hill Drive, Natick, MA, USA).

## 3. Results

According to the Kruskal–Wallis ANOVA, the WA group had a lower straight punch (Mean ± SD = 2156, *H* (2) = 35, *p* < 0.01, η^2^ = 0.52), palm strike (3445 ± 895 N, *H* (2) = 27, *p* < 0.01, η^2^ =0.38), and elbow strike (3158 ± 1217 N, *H* = 22, *p* < 0.01, η^2^ = 0.30) peak force than both other groups. Other groups’ differences were not statistically significant. 

The Friedman ANOVA showed that MP had a lower straight punch (Mean ± SD, 4662 ± 907 N, χ^2^_F_ (1) = 8, *p* < 0.01, *W* = 0.50) and elbow strike (5961 ± 2157 N, χ^2^_F_ (1) = 12, *p* < 0.01, *W* = 0.75) peak force in comparison to the palm strike (6460 ± 1612 N), and no differences between the straight punch and the elbow strike. MA had a lower peak force in the straight punch (4921 ± 1396 N) in comparison to the elbow strike (6047 ± 2482 N, χ^2^_F_ (1) = 13, *p* < 0.01, *W* = 0.22) and no differences in the palm strike (5512 ± 1638 N) to other types of strikes. The WA straight punch peak force was lower than the palm strike (χ^2^_F_ (1) = 22, *p* < 0.01, *W* = 0.42) and elbow strike (χ^2^_F_ (1) = 12, *p* < 0.01, *W* = 0.23) peak force. 

A maximum of three PCs were required in order to describe over 90% of the variance in the strike for all groups and conditions (Figure 2). The combination of the first three PCs explained more of the variance in the performance of the professional men for all three conditions, followed by the advanced women and then the advanced men (Figure 2).

The first principal component (PC1) explained more of the variance in the strike performance of the professional men than the advanced men or women for the punch and palm strike, but less of the variance for the elbow strike (Figure 3). The PC1 score curves were qualitatively similar for all groups for the palm strike, but differed for the other two strikes. In particular, there was more area underneath the PC1 score curve for advanced women in the punch and for professional men in the elbow strike (Figure 3). There was more divergence in the shape of the PC score curves for the second (PC2) and third principal component (PC3), although there were still substantial similarities. The greatest difference was seen in the PC2 curve of the professional men for the palm strike.

Peak force of the reconstructed strikes was highest in the palm strike for all groups (Figure 2) and the impulse expressed in the palm strike was greater than for the direct punch for the advanced groups (Table 1). In contrast, the punch impulse was greater than the palm strike impulse for the professional men. The largest total impulse was seen in the elbow strike for all groups.

## 4. Discussion

The aim of this study was to evaluate the time-series of net force of three strikes used in ‘ground and pound’ position across gender and performance levels. The elbow strike resulted in the highest reconstructed impulse and palm strike in the highest reconstructed peak force for all three groups, which can be explained by the characteristics of contact areas and different distances of the blows. The elbow is a more proximal segment than the hand, and thus requires more trunk support and a closer body distance and hip flexion during the contact phase. In addition, the distance required to reach the target is shorter than the other strikes and this can motivate the athlete to perform the strike more confidently. These factors are possible reasons for the higher impulse for the elbow strike than the hand strikes. Moreover, the elbow strike has a specific impact area—a sharp and small part of the humerus (olecranon) with a significantly smaller impact area than the palm strike and straight punch. It is also an exposed part of the forearms which can prolong the time for force interaction, because the imprint area can vary from a small size imprint of the olecranon to a large area of the forearm, due to the degree of flexion and pronation of the forearms [37]. Another factor that can cause differences between strikes is glove dependency. While the direct and palm strike are often performed with gloves as part of training, the elbow strike is performed without protection. This may play a role during the experiment, where athletes may be afraid to strike at full force without gloves. Martinez et al., 2018 [38] reported that a longer duration of the force curve and a lower peak created a higher impulse for leg bone loading. Thus, the same principle can be expected for bone tissue from a more general point of view, where Adamec et al., 2021 [39] indicates the small contact area, high rigidity of the impactor and the high impulse as a destructive type of impact. The higher peak force in the palm strike is in accordance with previous studies of Kung-fu strikes [11,39] and is probably explained by the tolerance of the palm impact area and also by the high frequency of this movement pattern in the daily locomotor needs of the upper limb. An open palm does not expose sharp and sensitive bone structures to the risk of injury or painful sensation, so athletes can invest more effort in the strike. Despite that the striking surface of the open palm is greater than a fist [40], a previous study reported that a palm strike can be performed with more emphasis on the wrist area and this lowers the area of (severe) effective mass and thus makes the palm strike more powerful [39]. For these reasons, the palm strike is generally recommended for beginners because execution options expand the tactical choice and make possible a more comfortable impact on the stable area comprising the scaphoid and pisiform bones. This should provide a better energy transfer [10] and thus a higher effect on the opponent’s body. We can confirm that all participants subjectively reported that the palm strike was more comfortable and produced a higher F_peak_ regardless of experience level than the direct punch and elbow strike. Moreover, in the professional group which has the best impact adaptation and technique [41], the highest F_peak_ in the palm strike was the most pronounced from all of the strikes, which was supported by both reconstructed and meaned values comparison. We are thus highly supportive of the recommendation of Bolander [10] who recommended the palm strike as the best strike to teach both novices and masters in self-defence, law enforcement, and emerging units due to its low variability and high F_peak_ values. 

The performance requirements of combat competitions clearly favour direct punching, as the straight punch is the most frequent of all strikes in combat sports [31]. Our results did not show a higher impulse or F_peak_ during the straight punch which has been reported in previous study [5] of ground and pound. Resultis in contrast with commonly held preconceptions about this strike [42] although the impulse was higher than in the palm strike in the professional population. Considering the small contact area of the straight punch [37,40] and the high rigidity of the metacarpal heads (contact area), this result is surprising. However, the magnitude of the direct punch net forces is what makes this strike dangerous for the opponent [39]. Due to the tactical aspects of the straight punch—i.e., that it is a “long distance strike”—we agree that this strike should be included more frequently in training than short-distance strikes [43,44].

PC1 explained between 70% and 84% of the variability in the impact force, and was similar for different genders and performance levels, which is in accordance with a previous study of impact values of different performance levels of boxers [45]. For the direct punch and palm strike, the contribution of PC1 was greater for the professional men than for the advanced athletes, which may be indicative of a more stable movement pattern. PC2, which includes the first harmonic of the impact peak, explained between 7% and 17% of the variability in impact force. Because the addition of the first harmonic to a curve can shift the peak of the curve, PC2 thus describes the variability in the timing of the main peak. It is notable that for the professional men, PC2 of the elbow strike was responsible for almost double the variability of PC2 for all other strikes and performance levels. This could be indicative of the increase in ‘good’ variability that is thought to be a characteristic of skilled performance. According to the uncontrolled manifold theory, the good variability is represented by significantly lower variability in orthogonal space (PC2), where elementary kinematic differences contribute to the stability of a strikes task [46]. Alternatively, it may indicate that a subset of the professional athletes were using a different technique, for instance, using their own effective body mass during the contact phase of the elbow strike. This is consistent with Lenetsky et al., 2015 [12], who identifies experience as the greatest predictor of using effective mass in a strike. 

It was common to see a double F_peak_ pattern in the raw data. This is exemplified by the pattern seen in the reconstructed curve for the direct punch. The second peak in the reconstructed curve for the direct peak in the direct punch occurs slightly after 0.01 s and corresponds with a peak seen in PC3 (Figure 3). This suggests that the variability in the biomodal pattern of impact force is largely captured by PC3; for the direct punch, the variability in the second peak is between 3% and 4%. A consideration of the curve for PC3 also suggests that a larger second peak would be associated with an earlier first peak. Details about double peaks can be supported by the result of an experiment that reported the movement of the centre of pressure at the exposed metacarpophalangeal joints after reaching the maximum impact force during a boxing straight punch [36].

The main study limitation is in analyses of the net dynamics only and not the strike movement characteristics such as body kinematics, strike velocity, and accuracy [47]. Another one is in the use of only the dominant side, which we expected to be preferred by athletes. The absence of hand protection was compensated by the foam covering force plate. However, the inelastic hardness may be a psychological barrier for the striker due to fear of injury. 

## 5. Conclusions

As all strikes permit the expression of high impact safely, all of them should be trained to provide a repertoire for tactical use. The palm strike should be included as it gives the highest F_peak_, which also increases with performance level. For closer distances, the elbow strike provides a good solution with a higher impulse than long distance strikes. The double force peak is a typical and significant characteristic of upper limb strikes, particularly for the direct punch.

## Figures and Tables

**Figure 1 sports-10-00205-f001:**
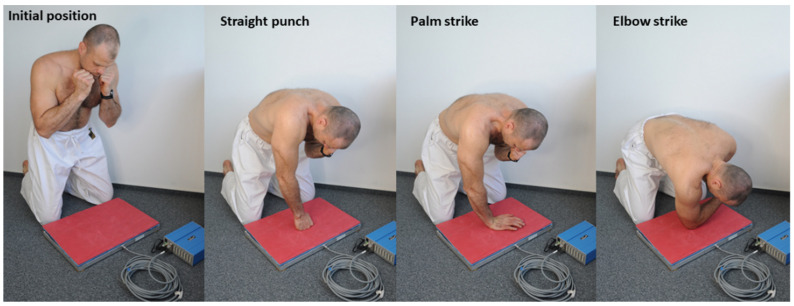
The initial and final body position for the straight punch, palm strike and elbow strike.

**Figure 2 sports-10-00205-f002:**
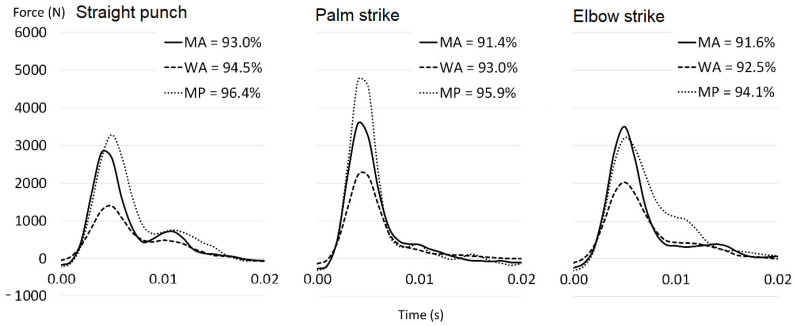
Reconstructed force time curves for direct punch, palm strike, and elbow strike of professional men (MP), advanced men (MA), and advanced women (WA). The force time curves are reconstructed by taking the linear sum of the first three principal components multiplied by the mean loading coefficient. The percentages give the variance explained by the first three principal components for each group and condition.

**Figure 3 sports-10-00205-f003:**
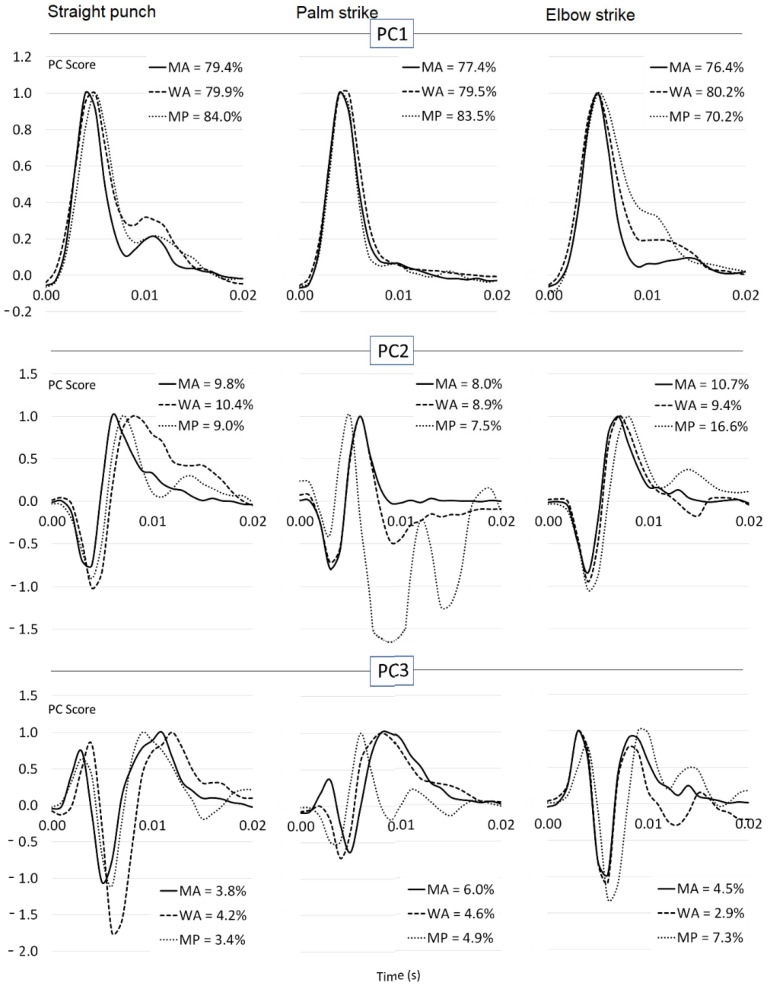
Principal component scores of the first three principal components for direct punch, palm strike and elbow strikes of professional men (MP), advanced men (MA) and advanced women (WA). The percentages describe the variance explained by each principal component.

**Table 1 sports-10-00205-t001:** Impulse (Ns) of reconstructed strikes by type of strike and performance level.

	Straight Punch	Palm Strike	Elbow
Advanced Men	13.5	13.9	15.5
Advanced Women	8.4	9.5	11.6
Professional Men	17.1	16.4	19.6

## Data Availability

The data presented in this study are available on request from the corresponding author. The data are not directly available due to their type and size.

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
