# Peer review of "Performance Level and Strike Type during Ground and Pound Determine Impact Characteristics and Net Force Variability"

_sports, 2022, doi:10.3390/sports10120205_

Round 1

Reviewer 1 Report

line 70 - Pay attention to: WÄ…sik J, Góra T. Impact of target selection on front kick kinematics in taekwondo – pilot study. Physical Activity Review 2016, 4:57-61. http://dx.doi.org/10.16926/par.2016.04.07

line 73 - "Therefore, the aim of this study was to compare ..." Maybe it's better to "evaluate"

line 91-95 How was the advancement of the participants rated ? Where was the line between professionals, advanced, etc.

line 179 - Are the differences between the groups statistically significant? I miss a statistical analysis, eg ANOVA.

line 188-189 Perhaps it is worth noting that the lower punch values may result from not preparing the straight punch surface, ie "hardening". MMA fighters use gloves, they may be afraid to strike at full force without gloves (?)

Author Response

line 70 - Pay attention to: WÄ…sik J, Góra T. Impact of target selection on front kick kinematics in taekwondo – pilot study. Physical Activity Review 2016, 4:57-61. http://dx.doi.org/10.16926/par.2016.04.07

Answer: Thank you for this suggestion. We have incorporated this into the discussion part.

line 73 - "Therefore, the aim of this study was to compare ..." Maybe it's better to "evaluate"

Answer: We agree and make this change in abstract and introduction.

line 91-95 How was the advancement of the participants rated ? Where was the line between professionals, advanced, etc.

Answer: Thank you for this point. We have clarified this point. We specified  it by the number of years for both expertise (and the instructor’s experience).

line 179 - Are the differences between the groups statistically significant? I miss a statistical analysis, eg ANOVA.

Answer: We agree, therefore we have added the comparative statistic for peak force values. However, please note that we consider the reconstructed curve as the major result and the comparison of the mean values as supportive. Therefore, we mainly commenting the reconstructed curve results in the discussion.  

line 188-189 Perhaps it is worth noting that the lower punch values may result from not preparing the straight punch surface, ie "hardening". MMA fighters use gloves, they may be afraid to strike at full force without gloves (?).

Answer: This was resolved by the protective math, which we stated in the “Instrumentation and data acquisition! Part. The relative dynamic attenuation was 28-42 % in peak force, but same for each participant. The mats attenuation has been added as a study limit also in limitation section. While the direct and palm strike are often performed with gloves as part of training, the elbow strike is performed without protection. This may play a role during the experiment, where athletes may be afraid to strike at full force without gloves.

Reviewer 2 Report

This manuscript entitled “Performance Level and Strike Type during Ground and Pound determine Impact characteristics and net force variability” primarily aimed to compare the net force variability of ground and pound strikes across genders and performance levels. The authors bring an interesting study, but there are still some problems that cannot up this review to a publishing level. Suggestions are listed in the specific comments below.

Specific comments:

1.     In the abstract part, line 18-19, “Eighty-one participants in three categories advanced, professional men and women…”, please provide detailed anthropometry information for participants, such as height, weight and body mass index.

2.     In the Materials and Methods part, participants, line 94-95, “Participants had been practicing self-defence at professional or advanced (> 5 years’ experience),” This sentence is unclear. Can you be more specific about how you define advanced and professional level?

3.     In the Materials and Methods part, participants, line 91-93, “The study sample consisted of 81 participants (men and women) in two performance categories, “it is recommended to add information about training experience.

4.     In the Materials and Methods part, Data curation and statistical approach, line 130-131, “The beginning of a strike was identified by finding the point at which the force increased to 300 N above the baseline force.” Why do you select 300N as a baseline?

5.     In the results part, figure 3, it is recommended to provide a new figure to replace the middle figure of PC2. The entire curve of MP is not complete.

6.     In the discussion part, it is recommended to provide a brief description of the aim and main findings in the first paragraph of the manuscript. Some recently studies could be added in the discussion, such as:

Upper Limb Strikes Reactive Forces in Mix Martial Art Athletes during Ground and Pound Tactics. Int. J. Environ. Res. Public Health 2020, 17, 7782. https://doi.org/10.3390/ijerph17217782

Development of Badminton-specific Footwork Training from Traditional Physical Exercise to Novel Intervention Approaches. Physical Activity and Health, 6(1), 219–225. DOI: http://doi.org/10.5334/paah.207

Lower Limb Biomechanics during the Topspin Forehand in Table Tennis: A Systemic Review. Bioengineering 2022, 9, 336. https://doi.org/10.3390/bioengineering9080336

7.     What are the limitations of this study? Please provide relevant description.

Author Response

This manuscript entitled “Performance Level and Strike Type during Ground and Pound determine Impact characteristics and net force variability” primarily aimed to compare the net force variability of ground and pound strikes across genders and performance levels. The authors bring an interesting study, but there are still some problems that cannot up this review to a publishing level. Suggestions are listed in the specific comments below.

Answer: Thank you for your detailed suggestions. We have added all details, which you suggested and incorporated some suggested references to the discussions.

 Specific comments:

  1. In the abstract part, line 18-19, “Eighty-one participants in three categories advanced, professional men and women…”, please provide detailed anthropometry information for participants, such as height, weight and body mass index.

Answer:  We have added this information to the abstract, due to the word count we have to reduce the first sentence before the study aim.

  1. In the Materials and Methods part, participants, line 94-95, “Participants had been practicing self-defence at professional or advanced (> 5 years’ experience),” This sentence is unclear. Can you be more specific about how you define advanced and professional level?

Answer: Thank you for the good notice. We specified the level of experience according to the number of practicing years now.

  1. In the Materials and Methods part, participants, line 91-93, “The study sample consisted of 81 participants (men and women) in two performance categories, “it is recommended to add information about training experience.

Answer: We have now specified the training experience.

  1. In the Materials and Methods part, Data curation and statistical approach, line 130-131, “The beginning of a strike was identified by finding the point at which the force increased to 300 N above the baseline force.” Why do you select 300N as a baseline?

Answer: The selection of 300N has been done because we observed many artifacts and shear forces, during initial contacts. The 300N was truly the first threshold that worked for all performance levels. This was due to the body support in some participants and the high slope of force gradient. We have tried lower the threshold but it would not sufficiently cover all strikes. This was also the reason why we did not compare impulse in comparative statistics, where our data cut might be the differentiating factor between strikes. On the other hand, this cut did not influence the peak force.  

  1. In the results part, figure 3, it is recommended to provide a new figure to replace the middle figure of PC2. The entire curve of MP is not complete.

Answer: Sorry for this flaw, we used the full curve figure now.

  1. In the discussion part, it is recommended to provide a brief description of the aim and main findings in the first paragraph of the manuscript. Some recently studies could be added in the discussion, such as:

Upper Limb Strikes Reactive Forces in Mix Martial Art Athletes during Ground and Pound Tactics. Int. J. Environ. Res. Public Health 2020, 17, 7782. https://doi.org/10.3390/ijerph17217782

Development of Badminton-specific Footwork Training from Traditional Physical Exercise to Novel Intervention Approaches. Physical Activity and Health, 6(1), 219–225. DOI: http://doi.org/10.5334/paah.207

Lower Limb Biomechanics during the Topspin Forehand in Table Tennis: A Systemic Review. Bioengineering 2022, 9, 336. https://doi.org/10.3390/bioengineering9080336

Answer: We have added the first reference to the discussion part and the aim at the beginning of the discussion, to introduce the main findings.

Thank you for  lower limb reports, but it does not include a description of an upper limb blows / strikes. Which we will take in consideration in our lower limb article.

  1. What are the limitations of this study? Please provide relevant description.

Answer: We have added the whole paragraph defining the major study limitations, into the discussion.

Reviewer 3 Report

I'm very sorry to write a negative review for this paper.

However, I  explain in order to pus the authors in improving their methological approach to research.

First, experimental set up. Does not have any meaning to test punches from a kneeling position.  The position simply does not exist and it is complete artefact not even comparable with a biomechanical similar movement.  At least the platform should be fixed on a wall.

The statistical approach (even if no information  is given about statistic, we can argue it was used matlab) is not correctly descibed.

To build representative curve does not have any meaning.

Several statements  (e.g. line 245 double peak)  and several inferences are not data based but descriptive and qualitative. 

The paper need a deep restructuring.

Author Response

I'm very sorry to write a negative review for this paper.

However, I  explain in order to pus the authors in improving their methodological approach to research.

Answer: Thank you for the time to read our manuscript.  We are surprised by your strictly negative feedback, which we tried to improve in our responses below. We believe that we resolve your major points.

First, experimental set up. Does not have any meaning to test punches from a kneeling position.  The position simply does not exist and it is complete artefact not even comparable with a biomechanical similar movement.  At least the platform should be fixed on a wall.

Answer: The kneeling position is one of the frequent end position in mix martial after take down and during the wrestling part (ground work) of the fight. This is already stated in the introduction in second paragraph with appropriate references. We have explicitly stated that MMA “mount position” actually mean our “kneeling position during ground”.

The second appearance is during self-defense, which is generally stated in the first paragraph.

Putting the desk on the wall is the aim of the different articles dealing with standing punches, not ground-and-pound tactics.

To address your point point: We added study (Miarka et al, 2016) at line 318 (references list) and at line 51, where groundwork techniques are described including the mount position with offensive techniques.

The statistical approach (even if no information  is given about statistic, we can argue it was used matlab) is not correctly descibed.

Answer: We agree, that we should explain the difference calculation more. Therefore, we have added peak differences calculation in the text. The impulse we have left only in reconstructed values, since the impulse has been cut at 300N, which might influence the comparative results (but not the reconstructed one).

To build representative curve does not have any meaning.

Several statements  (e.g. line 245 double peak)  and several inferences are not data based but descriptive and qualitative. 

Answer: Since our measurement is novel, we believe that our descriptive has a good value to future research. Especially when done on a very relevant sample size.

The paper need a deep restructuring

Answer: We better structured the result part. Otherwise, we would need a more detailed comment.

Round 2

Reviewer 2 Report

The authors have made a good revision, it is suitable to be accepted now. 

Author Response

Thank you for your positive evaluation.

Reviewer 3 Report

The paper has been improved. Although my  concern remains on using a kneeling position, the paper can be of interest for the readers

Author Response

Thank you for your positive evaluation, we understand your concern.